# Giant Viruses as a Source of Novel Enzymes for Biotechnological Application

**DOI:** 10.3390/pathogens11121453

**Published:** 2022-12-01

**Authors:** Ellen Gonçalves de Oliveira, João Victor Rodrigues Pessoa Carvalho, Bruna Barbosa Botelho, Clécio Alonso da Costa Filho, Lethícia Ribeiro Henriques, Bruna Luiza de Azevedo, Rodrigo Araújo Lima Rodrigues

**Affiliations:** 1Laboratório de Vírus, Departamento de Microbiologia, Universidade Federal de Minas Gerais, Belo Horizonte 31270-901, MG, Brazil; 2Núcleo de Apoio Técnico ao Ensino, Pesquisa e Extensão, Instituto de Ciências Ambientais, Químicas e Farmacêuticas, Universidade Federal de São Paulo, Diadema 09913-030, SP, Brazil

**Keywords:** giant viruses, nucleocytoviricota, NCLDV, phycodnaviridae, mimiviridae, genomics, enzymes, biotechnology

## Abstract

The global demand for industrial enzymes has been increasing in recent years, and the search for new sources of these biological products is intense, especially in microorganisms. Most known viruses have limited genetic machinery and, thus, have been overlooked by the enzyme industry for years. However, a peculiar group of viruses breaks this paradigm. Giant viruses of the phylum *Nucleocytoviricota* infect protists (i.e., algae and amoebae) and have complex genomes, reaching up to 2.7 Mb in length and encoding hundreds of genes. Different giant viruses have robust metabolic machinery, especially those in the *Phycodnaviridae* and *Mimiviridae* families. In this review, we present some peculiarities of giant viruses that infect protists and discuss why they should be seen as an outstanding source of new enzymes. We revisited the genomes of representatives of different groups of giant viruses and put together information about their enzymatic machinery, highlighting several genes to be explored in biotechnology involved in carbohydrate metabolism, DNA replication, and RNA processing, among others. Finally, we present additional evidence based on structural biology using chitinase as a model to reinforce the role of giant viruses as a source of novel enzymes for biotechnological application.

## 1. Introduction

The global demand and trade of industrial enzymes are continuously growing, and they are estimated to reach $7.0 billion USD in the next few years [1]. In this scenario, great importance is given to microbial enzymes, presenting several advantages, such as high yields, activity, and reproducibility, in addition to economic production, exponential growth, use of cheap platforms, and easy optimization. Many industrial processes demand enzymes, such as producing food, pharmaceutical products, detergents, and textiles. In this context, recombinant gene technology, protein engineering, and directed evolution have revolutionized enzyme manufacturing and this industry. Enzymes with a hydrolytic activity are used in the degradation processes of various natural substances and are extensively applied in industry. Proteases, essential enzymes for the detergent and dairy industries, are also widely used. Those enrolled in carbohydrate metabolism, including amylases and cellulases, are extensively used in the textile, detergent, and food industries [2]. Approximately 60% of industrial enzymes come from fungi, 24% from bacteria, 4% from yeasts, and most of the 12% remaining are obtained from plants and animals [1,2]. However, although viruses represent a small portion of these enzymes, studies about their potential have been growing in the last 30 years (Figure 1). They have a complex geometric structure and highly efficient genetic machinery, and, beyond that, they are sources of unique enzymes with great biotechnological potential [3].

The discovery of reverse transcriptase in retroviruses by David Baltimore and Howard Temin in 1970 was a milestone for molecular and cancer biology and started our understanding about retrovirology [4]. Other viruses that have been exploited for biotechnology are bacteriophages, since they are easy to manipulate and have interesting enzymes for several applications, such as DNA polymerases, DNA ligases, and lytic enzymes [2,5]. These lytic enzymes have enormous potential for use as antimicrobials because they exhibit bactericidal effects, absence of resistance, and activity against persistent cells. These enzymes degrade peptidoglycans, have antimicrobial and anti-biofilm properties (e.g., endolysins), and can be applied in treatments of bacterial infections [5,6].

The concept that viruses carry only genes that support their viral replication and capsid production changed with the discovery of giant viruses, opening space for a new approach and understanding of their contribution to the evolution of life [7]. Differently from the other viral groups and, similarly to bacteria and prokaryotes, they carry large genomes, with a diversity of genes capable of coding for numerous proteins, including DNA repair and even metabolic enzymes [7,8]. This new approach to understanding not only enriches the primary refinement regarding these viruses and their hosts, but also the beginning of the potential of these organisms for several biotechnological purposes. These viruses were first discovered in the 1970s, infecting unicellular algae, and many different isolates have been identified since then [9,10,11]. With the discovery of mimiviruses in the early 2000s and other giant amoeba viruses in the following years, the group of so-called nucleocytoplasmic large DNA viruses (NCLDV), currently classified in the phylum *Nucleocytoviricota*, greatly expanded, and pushed forward the boundaries of the virosphere [12,13,14].

Genomic studies of giant viruses of protists raised many questions about their biology, ecology, origin, and evolution. In addition, the surprising amount of genes harbored by these viruses, with a considerable number of them encoding enzymes used as valuable tools in different sectors of the economy, open a new venue for important novelties originated from viruses to be explored and applied in the biotechnology field [15,16,17]. In this review, we explore the diversity of giant viruses of protists, focusing on their genomic complexity, and highlighting how known genes previously described can be valuable sources of novel enzymes for biotechnological applications. 

## 2. *Phycodnaviridae*: The First Family of Giant Viruses of Protists

The *Phycodnaviridae* family includes viruses with biochemical and genetic peculiarities, such as DNA error correction and post-replicative processing, that infect eukaryotic algae from freshwater or marine environments [18,19]. Phylogenetic analysis using DNA polymerase B sequences from members of this family showed that they have a common ancestor with other NCLDV, thus corroborating their classification in the phylum *Nucleocytoviricota* [14]. The family currently comprises six genera named *Coccolithovirus*, *Phaeovirus*, *Prasinovirus*, *Raphidovirus*, *Prymnesiovirus*, and *Chlorovirus*, that differ in terms of cycle type, host, genome topology, and gene content [20,21].

Although genes enrolled in lipid metabolism are not the most abundant functional category in giant viruses, they are strongly present in coccolithoviruses (Figure 2). Coccolithoviruses infect microalgae of the species *Emiliania huxleyi*, commonly found in marine sediments. Emiliania huxleyi virus 86 (EhV-86), one the first known coccolithoviruses, was observed in 1999, having a viral particle around 200 nm, covered by a lipid membrane and a linear genome of 407 kbp [22]. Curiously, a genomic characterization of the EhV-86 identified 472 coding sequencing (CDS) regions, but only 63 genes have a known function so far (Table 1). An amount of 10% of them encode enzymes involved in the biosynthesis of sphingolipids: sterol desaturase, serine palmitoyltransferase, lipid phosphate phosphatase, and two genes encoding desaturases (Figure 2; Appendix A).

Such enzymes are involved in the synthesis of ceramide, which induces apoptosis of the infected cell [23]. Although the mechanism of apoptosis has already been observed in other viruses, no genes related to the synthesis of sphingolipids were found in their genomes, making these genes exclusive to coccolithoviruses [24,25,26]. In addition, a proteome analysis showed that these enzymes enrolled in sphingolipid biosynthesis are present as early-class proteins, suggesting that they could be functional and also play an important role in initial infection [27]. Besides that, this highlights that these genes are not only carried inside the viral particles, but that they are also translated into functional proteins in the host and can be explored as biotechnological enzymes. Sphingolipids are molecules found in eukaryotes and prokaryotes and perform structural, signaling, and biochemical functions. They have been mentioned as a potent food supplement, and as a cosmetic, as they prevent skin infection and inhibit bacteria and fungi proliferation [28,29].

A comparative study performed by Nissimov and colleagues showed the presence of 25 to 29 CDS from other isolated viruses (EhV-201, EhV-207, and EhV-208) identical to sequences present in the EhV-86 genome. The predicted enzymes found were methyltransferases, glycosyltransferases, and RNase, and the majority of non-shared proteins, considered hypothetical ones, have unknown functions. On the other hand, the EhV-84 isolate showed many proteins (231 CDS) with identical homology with EhV-86 proteins [30]. With a few available genomes, it is clear that there is a vast field to be explored, both for obtaining more information about the viruses’ biology and ecology, and to be investigated for biotechnological purposes.

Phaeoviruses infect filamentous algae, most from the genus Ectocarpus and Feldmania, in subtropical environments, and they are the only phycodnaviruses known so far to infect more than one host. Genomic analysis of the Ectocarpus siliculosus virus-1 (EsV-1) revealed 231 CDS regions, where only 50% had determined functional characterization. Among these include genes involved in DNA synthesis, polysaccharide metabolism, histidine protein kinases, integration, and transposition [31]. Integrases catalyze site-specific DNA rearrangement, and transposases can bind in transposons on DNA and move small fragments along the genome. Both enzymes can be used for gene editing, gene therapy, and integrases are also studied as resistant markers [32,33]. A close relative is the Feldmania species virus (FsV), a phaeovirus associated with the brown filamentous algae *Feldmania* sp. This virus was considered the smallest giant viruses with a linear genome of 154 kbp and 150 CDS regions, of which only 25% had similarity with the database, such as those enrolled on DNA replication, transcription, nucleotide metabolism, and also lipid and protein metabolisms (Figure 2 and Figure 3) [34].

Prasinoviruses infect prasinophytes, considered the smallest free-living photosynthetic eukaryotes [35]. Genomic analysis of the Osteococcus tauri virus (OtV-1) showed 232 CDS regions, where 31% showed functional similarity with previously described proteins, including methyltransferases and other enzymes involved in DNA, protein, and carbohydrate metabolism [33]. The Osteococcus tauri virus OtV-5 genome has 268 CDSs, and only 57% of the predicted proteins had a known function, including those involved in DNA replication and viral particle formation. Interestingly, some host-related genes were also found, including a proline dehydrogenase, related to cellular oxidation protective metabolism [36,37]. This virus has complex glycosylation machinery, with at least five glycosyltransferases and a galactosyltransferase, indicating relative independence of the host for glycosylating their own proteins (Figure 2; Appendix A). It’s worth noting that other giant viruses also have glycosylation machinery, with many proteins involved in carbohydrate modification and sugar production, which could be explored in the biotechnology industry [38,39].

Rhaphidoviruses have a wide variety of hosts. Among them is the single-celled seaweed bloom-forming *Heterosigma akashiwo*, which can form surface aggregations toxic to the environment [40]. The complete sequencing of the first virus strain infecting this alga (HaV53) was published in 2016, and genes related to DNA regulation, carbohydrate metabolism, signal transduction, and regulation of ubiquitin-related proteins were found. However, there is still a limited characterization of this genome [41]. Similar to other members of the *Phycodnaviridae* family, HaV01 has known glycosyltransferases that might be involved in viral protein glycosylation. In addition, proteins involved in transcription and RNA processing have also been identified, including a ribonuclease III and a mRNA-capping enzyme (Figure 2; Appendix A). Ribonuclease III can cleave double-strand RNA (dsRNA), an essential step in the maturation and decay of coding and non-coding RNAs. The first characterized ribonuclease III was from *Escherichia coli*, which is commercially available, and the enzyme is also present and well-conserved in plants, animals, fungi, and eukaryotic viruses [42]. The mRNA-capping enzyme is a complex that promotes the first modification of RNA polymerase II transcripts. In this context, this complex can regulate cap-dependent protein synthesis and act in the protein export mechanism [43]. Many types of mRNA-capping systems have also been described in viruses, such as influenza, orthomyxoviruses, alphaviruses, mimiviruses, and chloroviruses [43,44]. It is interesting to note that New England Biolabs Inc. has recently announced that the Faustovirus capping enzyme is commercially available, an enzyme that demonstrates increased capping efficiency across a variety of mRNA 5′ structures than previous enzymes [45].

Prymnesioviruses infect phytoplankton algae with high biomass formation, such as *Phaeocystis globosa*. Genomic analysis of the strain Phaetocistis globosa virus-16T (PgV-16T) showed 434 CDS regions with no phylogenetic proximity with the other viruses that infect algae, even though they are part of the Megaviridae clade. Seventy percent of its genome is similar to other large double-stranded DNA (dsDNA) viruses, with genes related to many processes, such as DNA replication and repair, including methyltransferases and transposases [46]. Seven coded genes seem unique in their genome among the group, which encode peculiar enzymes, such as phospholipase and asparagine synthetase homologs [46]. Phospholipases are responsible for hydrolyzing phospholipids into other lipids and are widely used in industrial food processes, while asparagine synthetase is a target related to the growth of human tumor cells. These prokaryote enzymes have also been characterized [47,48]. Compared to other phycodnaviruses, the difference between PgV-16T and these viruses’ genetic profile is clear, considering the functional clusters of genes (Figure 2). Such a difference corroborates previous data, pointing to this virus as a member of the *Mimiviridae* family [46]. Another member of this group is the Chrysochromulina brevifilum virus PW1, the only recognized species by ICTV so far [20]. A few viruses infecting *Chrysochromulina* sp. have been identified in the last years, and genome analysis of C. parva viruses suggested limited gene machinery compared to other phycodnaviruses (Figure 3). 

The last-mentioned genus, *Chlorovirus*, was the first to be created, comprising the first virus associated with chlorella-like green algae, back in the late 1970s [49,50]. The first reported phycodnavirus, zoochlorella cell virus (ZCV), was isolated in the late 1970s in Japan from a *Chlorella* sp. that lives in symbiosis with the protozoan Paramecium bursaria. The ZCV was able to infect only zoochlorella recently separated from its symbiotic protozoan [9]. A few years later, viruses sharing many characteristics with ZCV were isolated from zoochlorella associated with *Hydra viridis* (HVCV-1 and HVCV-2) and also with *Paramecium bursaria* (PBCV-1), which would become the most studied algae viruses over the last decades [10,49,50]. Given the importance of chloroviruses for the field, we will cover the main genomic features of these viruses separately.

## 3. Chloroviruses: Large Viruses with Giant and Complex Genomes

Representatives of the *Chlorovirus* genus have a complex linear genome ranging from 290 kb to 370 kb and containing 315 to 415 protein-coding sequences. Approximately fifty percent of their CDS have no known function, and some have never been observed in other viral genomes [21]. 

A high tropism observed between choloviruses and chlorella cells is probably related to the viral interaction with the host’s cell wall, a thick and rigid structure that varies according to the alga species [51]. The chloroviruses isolated from *Chlorella variabilis* NC64A are the NC64A viruses; those that infect *Chlorella heliozoae* SAG 3.83 are the SAG viruses, and the Pbi viruses have the alga *Micractinium conductrix* Pbi as an in vitro host. Those that replicate in vitro exclusively in *Chlorella variabilis* Syngen 2-3 are named Osy viruses [52].

Gene organization is relatively conserved within the same clade, even though synteny is not well conserved in chloroviruses. Genes related to the initial and final phases of the infection are dispersed throughout the genome, but gene clusters are observed as “gene gangs” [21,53]. There are groups of orthologous genes (COGs), specific for each of the three main host-related chlorovirus clades (NC64A, Pbi, and SAG), which suggests that these genes may encode proteins related to tropism and recognition of cellular receptors of the microalgae host [54]. 

The 331 kb genome of PBCV-1, the study model of chloroviruses, has repeated inversions covalently linked that form hairpins at the far ends [55]. The approximately 130 protein-coding sequences common to all members of the genus constitute the core genome, composed of just over 45% of coding sequences related to proteins with known functions [54,56]. These proteins are related to viral multiplication and essential functions common to all chloroviruses, such as viral DNA replication, virion structure, and host cell wall degradation [54]. Intronic regions (up to three types of introns) have also been identified in the chlorovirus genome, and some are highly conserved within the genus [11]. Viral DNA has methylated bases that occur at highly variable frequencies, even comparing genomes of the same viral species, and it is resistant to degradation by endonucleases [19].

Enzymes with unique characteristics have been previously found in chloroviruses and are commercially available as biotechnological products. The PBCV-1 DNA ligase (SplintR Ligase), considered the smallest known functional ligase, can establish a high-efficiency binding between a DNA molecule and a target RNA molecule, allowing the construction of RNAseq libraries and microRNA studies [57,58]. The CvIAII endonuclease was identified in the 1990s, and this enzyme cleaves DNA at the specific C/ATG site and is not affected by mATG methylation. Both enzymes are produced exclusively by these viruses and are commercially available by New England BioLabs Inc. [59]. Furthermore, chloroviruses are rich in methyltransferase systems and have exclusive enzymes involved in carbohydrate metabolism, which give these organisms a unique biotechnological potential [60,61,62,63].

Chloroviruses have many genes encoding enzymes of carbohydrate metabolism (Figure 2; Appendix A). In PBCV-1, UDP-glucose dehydrogenase (UDP-GlcDH), fructose-6-phosphate-aminotransferase, and hyaluronan synthase were identified, and their transcripts observed in the initial phase of infection in *Chlorella* sp. [60,61]. These genes are involved in the synthesis of hyaluronan, a polysaccharide produced in the dense host cell, forming a fiber network [64]. Hyaluronan or hyaluronic acid is a polymer of repeated units of β-1,4-D-glucuronic acid and β-1,3-N-acetyl-D-glucosamine, a component of extracellular matrices in mammals [65]. It is applied in several areas of the biotechnology industry, from immunomodulation and tissue regeneration to the production of nutraceutical cosmetics [66,67,68]. The *has* gene codes for hyaluronan synthase and is found in only 30% of chloroviruses. This gene does not appear to be essential for viral replication, either in the laboratory or under natural conditions [11,64]. However, it was demonstrated that the production of hialuronan in extracellular matrix of chlorella after infection with chloroviruses is observed. Although the role of polysaccharide production by infected cells, and why chloroviruses carry these genes is still unclear, one can consider that these carbohydrates play essential roles in their multiplication cycle [11].

Other essential enzymes are chitinases, chitosanases, β-1,3 glucanase, and alginate lyases, which seem to participate in the degradation of host cell wall polysaccharides, apparently associated with initial infection [69]. Although chitin is not commonly present in the cell wall of green algae, it was found in algae after Pbi or CVK-2 chloroviruses infection, suggesting that its synthesis was likely a result of chloroviruses infection [70,71]. Thus, despite the mechanism by which the synthesis and degradation of these carbohydrates occur in these organisms is not yet elucidated, the host–virus interaction is essential to produce these proteins.

Chitinases catalyze chitin hydrolysis reaction, while chitosanases act the same way on chitosan, producing molecules with lower molecular weight used in the production of polymers and biofilms and biological controls [72,73,74]. In agriculture, the inhibitory properties of chitinases have been used for the biological control of plant pests and fungal diseases [75,76]. They are also used for biomass degradation of chitin into chitooligosaccharides (COS), chitosan, and other chitin derivates up to the production of soluble monomers that can be used in the food industry [73,77,78]. In the same way, oligosaccharides resulting from chitin degradation have also demonstrated antitumor and anti-inflammatory potential [79,80]. 

Interestingly, chitinase of PBCV-1 (*Phycodnaviridae*), Tupanvirus (*Mimiviridae*), and *Bacillus cereus* (Bacteria/Bacillaceae) share a conserved sequence D/SI/LDWEY, which could correspond to the aspartate and glutamic acid catalytic residues in common chitinases (e.g., D209 and E211 in PDB ID: 6BT9), even with an evident difference in the size of the proteins (Figure 4, Appendix A). Target sequences were also modeled with RoseTTAFold, presenting similar results to SWISS-MODEL predictions, showing an equivalent core for all structures, evident differences in size, and also two lateral protein fragments for *B. cereus*, one for Tupanvirus, and being absent in PBCV-1 (Appendix A, Appendix A). Chitinase of *B. cereus* has been studied as a promising molecule for biological control with a potential antifungal activity [81]. The similarity found in the catalytic domain of the three proteins’ structure is additional evidence that giant viruses can share functional enzymes with prokaryotes and reinforces that, as well as bacteria (which represent one of the most representative groups for enzymes with biotechnological potential), they can also be regarded as a rich source of functional enzymes. Corroborating our data, a recent study also demonstrates that chitinase encoded by a virus genome is active, as well as bacteria showing the insecticidal effect [82].

Another complex machinery is the addition and removal of carbohydrates, a process called glycosylation, which promotes the modification of proteins that can be connected to the cell, including the conversion of signal transmission, molecular signal, and endocytosis, key events that confer structural diversity among organisms [83,84]. The PBCV-1 genome has glycosyltransferases, GDP-d-mannose dehydratase (GMD), and GDP-4-keto-6-deoxy-d-mannose epimerase/reductase (GMER) enzymes. These are enrolled in the glycosylation process, synthesizing the sugar residues fucose, rhamnose, and glycans, constituents of the main capsid protein Vp54 [83,85,86]. They are highly conserved enzymes distributed in cellular organisms, involved in the formation of GDP-L-fucose in bacteria, plants, and animals [85,86,87,88,89].

Glycosyltransferases can transfer sugars for different biomolecules, such as lipids and peptides. They play an essential role and have been studied as a tool to develop new drugs, vaccines, and therapeutics [90]. They also can be used to convert and obtain glycosides with biological interest, such as a rapid conversion of sucrose and uridine 5′-diphosphate (UDP) into UDP-glucose at large scale in different organisms [91,92]. Another essential function is the post-translational modification in proteins, recognized as N-glycosylation, which is found in many biopharmaceutical proteins and can influence their solubility, functionality, and other properties. In this context, modifications in the N-glycosylation process can be achieved to obtain products with different characteristics [93]. Altogether, chloroviruses and other phycodnaviruses have several genes to be further structurally and functionally characterized, which could bring exciting and innovative biotechnology tools. After the discovery and expansion of the viruses infecting photosynthetic protists, new viruses were identified through infecting free-living amoebae. Another group of protists that have been hiding in plain sight are a valuable source of giant and complex viruses [94].

## 4. Giant Viruses of Amoebae: Expanding the Complexity of the Virosphere

The algae-infecting *Phycodnaviridae* were the first family of viruses referred to as giants [95]. However, discussion and further studies of giant viruses greatly intensified with the discovery of Acanthamoeba polyphaga mimivirus (APMV) in 2003, a virus that can infect free-living amoebae of the *Acanthamoeba* genus [12]. Mimiviruses form a broad and diverse family of viruses belonging to the phylum *Nucleocytoviricota*. According to the International Committee on Taxonomy of Viruses (ICTV), the *Mimiviridae* family is currently composed of two recognized genera: *Cafeteriavirus* and *Mimivirus*, whose main representative members are Cafeteria roenbergensis virus (CroV) and APMV, respectively [20]. CroV is known to infect a marine heterotrophic unicellular protist named *Cafeteria roenbergensis* and has a genome of 730 kbp [96]. On the other hand, APMV harboring its 1.2 megabase pairs (Mbp) genome is known to use amoebas from the *Acanthamoeba* genus as hosts, at least in laboratory conditions (Table 1) [12]. Although ICTV currently classifies only a couple of taxonomic groups belonging to the *Mimiviridae* family, several other putative members of this family have been described in the last years. Many isolates have already been identified in environmental water samples in different countries in Oceania, Europe, Asia, Africa, and South America [97,98,99,100,101]. APMV is the main representative member of lineage A, whereas moumouviruses represent lineage B, and megaviruses represent lineage C [102,103,104]. 

The discovery of these viruses greatly impacted the virology field due to the particles and genome sizes of mimiviruses composed of unique gene machinery. In general, mimiviruses’ dsDNA molecules code for genes never described for any virus before. This set of genes includes some related to protein translation and DNA repair processes, as well as chaperones and genes involved in different enzymatic pathways [13,105]. The mimiviruses translation-related genes set includes many aminoacyl-tRNA synthetases, transfer RNAs (tRNAs), and translation factors [105,106,107]. AARSs can establish a covalent ligation between an amino acid and its cognation tRNA and are important for metabolic and signaling pathways. In that way, because they are enrolled in protein translation, in biotechnology they are important to study protein regulation and are also an interesting target for drug discoveries [108,109]. In addition, the category of enzymatic pathways-related genes comprises enzymes involved in amino acid and lipid synthesis, sugar metabolism, and protein glycosylation, similar to that described for phycodnaviruses. APMV, for example, codes for at least six classes of glycosyltransferases that might be involved in the glycosylation process of its major capsid protein (MCP) and its fibrils’ glycoproteins [105,110].

Besides the three lineages, more divergent members of the *Mimiviridae* family were also described recently. The tupanviruses are intriguing giant viruses isolated from extreme environments in Brazil, having a capsid-associated long tail, which enables the formation of gigantic particles (~2.3 µm) [111]. Tupanviruses also drew attention because of the complexity of their genomes. These viruses have the most complete protein translation apparatus of the virosphere to date, which includes up to 70 types of tRNAs, and factors related to tRNA maturation and stabilization besides all the 20 aminoacyl-tRNA synthetases [111]. Interestingly, a peculiarity of tupanviruses is that they code for citrate synthase, an enzyme involved in the metabolic pathway of energy production [112]. This enzyme is essential for starting the tricarboxylic acid cycle in eukaryotes and prokaryotes, which is an important key for cell energy production [113]. It has been used in the biotechnology industry to measure pyruvate carboxylase activity by enzymatic assay tests and also as a mitochondrial biomarker in cells [114]). Additionally, tupanviruses encode a mannose-specific lectin gene, which seems to be related to the amoebal-bunch formation, a specific cytopathic effect caused by these viruses in amoebas [115]. In general, lectins bind to different carbohydrates, and some recent studies have shown their potential biotechnological roles, such as the purification of biomolecules and insecticide action [116]. Tupanviruses, as well as mimiviruses, have complex and quasi-autonomous glycosylation machinery [38,39]. These viruses have many enzymes involved in carbohydrate metabolism, including glycosyltransferases, glucose-methanol-choline oxidoreductases, and UDP-glucose 4 epimerase, among others, accounting for 3% of viral genomes (Figure 3). Curiously, Tupanvirus soda lake has a chitinase coded in its genome, which is homologous to a chitinase found in chloroviruses (Figure 4). Both viral enzymes have structural similarities with a chitinase from *Bacillus cereus* (a microbial source of industrial enzymes), including specific residues at the enzyme active site (Figure 4). The biological activity of this protein remains to be characterized, but evidence points to these viruses as promising sources of new active enzymes. 

Some mimiviruses that infect algae can also have exclusive and intriguing genes. Tetraselmis virus (TetV-1), a mimivirus that infects green algae, is the only virus of the family having fermentation genes, such as those enrolled in pyruvate metabolism, pyruvate formate-lyase, pyruvate formate-lyase activating enzyme, mannitol metabolism, and mannitol 1-phosphate dehydrogenase. They also have alpha-galactosidases, genes enrolled in sugar degradation [117]. Fermentation enzymes are related to glycolysis anaerobic metabolism and are commonly found in bacteria. Because of them, these organisms are widely used in industry to produce ethanol, food, and medicines [118]. Numerous genes encoding enzymes enrolled on triacylglycerol degradation were also found in Prymnesium kappa virus RF01 (PkV RF01), which also infects algae [8,119]. Lipase enzymes catalyze the hydrolysis of triglycerides to glycerol and fatty acids. These enzymes have potential applications in oil, food, biodiesel production, and many other industries. Recently, they have been studied as a potent tool to nutritionally enrich vegetable oils or remove phospholipids, which are unwanted molecules [120,121]. 

Another group of viruses related to mimiviruses was first described through metagenomics approaches, the so-called klosneuviruses. Similar to tupanviruses, the klosneuviruses possess an extensive protein translation apparatus and comprise a distinct clade within *Mimiviridae* [122]. In recent years, a few viruses have been isolated, corroborating the existence of the putative Klosneuviridae subfamily with highly complex genomes reaching up to 2.0 Mb and over 1000 genes (Table 1) [123,124]. Members of the Klosneuvirinae group have many genes associated with distinct functions (Figure 3). Similar to other mimiviruses, the klosneuviruses have many glycosyltransferases, composing the glycosylation machinery of these giant viruses. For instance, Fadolivirus has at least 20 glycosyltransferases, one of the most complex glycosylation apparatuses observed in a virus. Additionally, a robust apparatus for DNA replication and repair are found, including several endonucleases, topoisomerases, and helicases. As for other giants, most of the genes coding for klosneuviruses remain to be characterized, and in-depth investigation might reveal plenty of new enzymes applicable in biotechnology.

Besides the *Mimiviridae* family, several other groups of large and giant viruses of amoeba are proposed to be part of the *Nucleocytoviricota* phylum, such as Marseilleviruses, pandoraviruses, molliviruses, pithoviruses, cedratviruses, Faustoviruses, kaumoebaviruses, and orpheoviruses, among others [14]. In 2009, the first Marseillevirus isolate was found in a water sample from a Paris cooling tower, characterizing the second and widely distributed new family of NCLDV viruses that infect amoebas [125]. From there, new viruses were discovered in samples of water, soil, mussels, and even humans, totaling more than 50 isolates found in different places worldwide, including Europe, Africa, America, Oceania, and Asia. Unlike mimiviruses, the Marseilleviruses do not have a robust translation-related gene set, but code for different types of histone-like proteins, a remarkable characteristic of the family [126,127]. More recently, other viruses isolated in France (Clandestino virus) and Japan (Medusavirus) expanded the histone-like proteins in the virosphere [128]. It was shown that the doublet histones are essential for Marseillevirus infectivity and that they form nucleosome-like structures, thus analogously organizing the viral genome as eukaryotes [129]. Additionally, Marseillevirus genomes present a high rate of mosaicism since the genes have different putative origins, such as other viruses, bacteria, archaea, and eukaryotes [125,130]. Similar to phycodnaviruses, some Marseilleviruses and other giant viruses of amoebae have a complex restriction–modification system composed of different methyltransferases, suggesting the involvement in diverse forms of virus-host interactions [63]. These enzymes are important tools in biotechnology, applied to facilitate DNA-based genetic engineering [131]. Additionally, interesting as molecular biology tools, Marseilleviruses encode many proteins involved in RNA processing, such as RNAse H, mRNA capping enzyme, and RNA ligase. There is still no information about how these enzymes work in the virus cycle, but further characterization can lead to exciting discoveries for both virus biology and biotechnology.

The first members of the putative family Pandoraviridae were described in 2013. Pandoravirus salinus was isolated from samples collected on the central coast of Chile, with a genome size of 2.5 Mb, and Pandoravirus dulcis in a freshwater lagoon in Melbourne (Australia), with a genome of 1.9 Mb [132]. Compared to mimiviruses, they have a giant genome with 93% non-recognizable homologs, while mimiviruses have around 50% [105,132]. This giant genome harbors genes involved in functions that raise many questions about the nature of viruses. Unlike other nucleocytoviruses, the pandoraviruses do not code for the typical double-jelly-roll capsid protein. Moreover, a recent study suggested that pandoraviruses code for homologs of enzymes involved in the tricarboxylic acid cycle (TCA), including a functional isocitrate dehydrogenase. The TCA is related to acetyl-CoA oxidation to produce energy in cellular organisms. Interestingly, these pandoraviruses’ putative enzymes are transcribed during the pandoravirus massiliensis replication cycle, highlighting their importance during the infection [133].

In this context, viruses can be a passive vehicle for transporting genes through host cells, but they also participate in recycling nutrients to these hosts in their natural environment [134]. Notably, some phage genes are functional in cyanobacteria photosynthesis during infection and can act as supplement proteins in these organisms [135]. In the same way, genes enrolled in cell energy production present in giant viruses, such as pandoraviruses, can enhance physiological machinery present in the host cells with biotechnological interest, such as those enrolled in photosynthesis [133]. Horizontal gene transfer between giant viruses and their hosts is widely discussed, especially considering those that infect algae. This event can explain how they carry peculiar genes in their genomes that act on viral DNA replication or protein synthesis, in addition to enhancing the physiological potential of their hosts [136]. This is similar to the presence of unexpected genes enrolled in cellular redox potential, including thioredoxin family protein, thiol oxidoreductase, and ferric reductase. They also have transporters for inorganic ions, such as ammonium, magnesium, and phosphate. These proteins are commonly found in algae and are vital for viral and cell host injury survival, but they can also contribute to modifying the chemical composition of the environment, e.g., ferric reductases can facilitate iron uptake [8,137,138,139]. This mechanism in chlorella algae is widely studied since these organisms are natural iron chelators, but the reason why viruses that infect these algae carry those genes remains curious [140,141].

Other groups include icosahedral viruses closely related to *Asfarviridae*, the Faustoviruses, kaumoebaviruses, and pacmanvirus [142,143,144]. Faustoviruses were first reported in 2015, with eight distinct strains infecting *Vermamoeba vermiformis*, a protozoan associated with human environments. They are icosahedral-shaped viruses of around 200 nm and have a genome size of about 466 kbp (Table 1) [142]. These viruses have many different enzymes. Recently the release of the Faustovirus Capping Enzyme (FCE), an enzyme combining high activity and a broad temperature range applicable in mRNA manufacturing, was announced, corroborating the importance of giant viruses as a source of new enzymes [45]. Pacmanvirus A23 and kaumoebavirus also have mRNA capping enzymes, and despite low identity (~30%), it is possible that such homologs are also active and applicable in further assays. In addition, these viruses have DNA ligases and restriction endonucleases, enzymes extensively used as molecular biology tools. Additionally, they have different proteases (e.g., serine and cysteine protease). Proteases have long been used in biotechnology and industries, with applications in various processes such as detergent, textile, leather, and dairy products [2]. These proteins usually come from fungi and bacteria, and now viruses appear as a promising source of new proteases.

Finally, a group of ellipsoid viruses exhibits peculiar features, with giant particles and relatively small circular DNA genomes, composing a putative family Pithoviridae. The first pithovirus was isolated from Siberian permafrost samples dated 30,000-years-old [145]. A few years later, a contemporary relative was isolated in France, and more recently, metagenomics studies found several genomes similar to pithoviruses expanding the new group [146,147]. These viruses have the largest particles ever described, with a mean value of 1.5 µm and dsDNA genomes of ~610 kbp. As for other giant viruses, most of the genes coded by pithoviruses are yet to be characterized, but there are some interesting genes for possible application in biotechnology, including serine protease, methyltransferase, DNA ligase, and nucleases (both DNAse and RNAse). Cedratviruses have a similar enzymatic profile. These viruses form a sister clade of pithoviruses, having similar ellipsoid particles of ~1.0 µm and circular dsDNA genomes ranging from 460–590 kbp [131]. It is interesting to note that giant viruses have active methyltransferases and nucleases, constituting a viral restriction-modification system [63]. These systems could be further exploited for exogenous gene expression regulation and DNA plasmid stability as promising biotechnology tools [148]. Completing this putative new taxon is Orpheovirus, an oval-shaped virus of 900–1300 nm with circular dsDNA of 1.4 Mb [148]. Despite some similarities with pithoviruses, including viral particle and replication cycle [149], there are considerable genomic differences (e.g., the presence of translational-related genes), and phylogenetic analysis put Orpheovirus as a distant relative of pithoviruses, possibly inaugurating a new viral family Orpheoviridae. There is still limited information about these viruses, with only one viral isolate and a few genomes found by metagenomics [148,150]. Yet, it is interesting to note that these viruses have robust enzymatic machinery of interest for application in biotechnology, composed of nucleases, ligases, helicases, and lipases.

## 5. Conclusions and Future Directions

Giant viruses have genetic, proteomic, and structural complexities unique to the virosphere. Their large genome carries not only structural genes or elements necessary for DNA replication, but also genes of complex machinery common to organisms, such as bacteria, archaea, and small eukaryotes. It includes methyltransferase and glycosylation systems, as well as several enzymes involved in protein, carbohydrates, and polysaccharides metabolism. In addition to a vast genetic repertoire, viral particles harbor mRNAs that can encode different proteins [151,152]. Although most of their genomes are still unknown, studies indicate a correlation between the discovery of new genes and the prospection and isolation of new viruses, showing an open pan-genome in these viruses [56,153]. One can expect that, with the isolation of more viruses from unexplored regions of the Earth, it will be possible to identify a plethora of new genes with the most diverse functional activities that could be explored through the lens of biotechnology. In fact, a few years ago, a completely new amoeba-infecting virus was isolated and characterized in Brazil: the Yaravirus. This curious small virus (80 nm) that appears among the amoeba-infecting giants has around 90% of its genome composed of ORFans, and further characterization might reveal important novel elements essential for the virus replication and possible application in biotechnology [154]. Due to its remarkable and different features, yaravirus is currently classified into the *Yaraviridae* family by ICTV [155]. Together, all these genomic features of amoebal and algae viruses highlight their potential as biotechnological tools.

Metagenomics studies suggest that viral particles are more prevalent in aquatic environments than bacteria and that phycodnaviruses constitute one of the most abundant viral groups in the ocean. Metagenomics approaches deeply impacted and expanded the field of giant viruses in recent years (for a more extensive review, please see [15]). Using this strategy, a plethora of new genomes was recovered from different places around the world, dramatically improving the phylogenetic diversity of these viruses and providing important insights into virus-host interactions [137,156]. Additionally, these studies provided information on the dynamics of genome evolution, revealing many new protein-coding genes that could be further explored through the lens of biotechnology. Metagenomics studies performed in forest soils and deep-sea sediments have uncovered a hidden diversity of giant viruses in this environment, evidencing that they are really ubiquitous in our planet [157,158].

Many studies argue that aquatic viruses represent the greatest unexplored genetic diversity on the planet [16,159,160,161,162]. It is worth noting that such diversity will likely improve in the years to come due to climate change, allowing new discoveries. Recent metagenomics studies in permafrost led to the discovery of new giant viruses, constituting a large reservoir of genes of unknown function [150]. A few giant viruses have already been recovered from Siberian permafrosts, raising the question of what else could be brought to the surface with climate change [145,163]. Exploring such genetic richness will bring exciting innovations that can be the key to different problems we are dealing with and might come across in the biotechnological field. Using new approaches for better characterizing the giant viruses’ genomes with new algorithms by applying artificial intelligence (e.g., AlphaFold) and further advancing with the biological characterization of new proteins is the next step of possible scientific breakthroughs. Giant viruses are everywhere, just waiting for scientists to face the challenge of unraveling their mysteries and finding innovative ways to use their complex enzymatic machinery to improve science.

## Figures and Tables

**Figure 1 pathogens-11-01453-f001:**
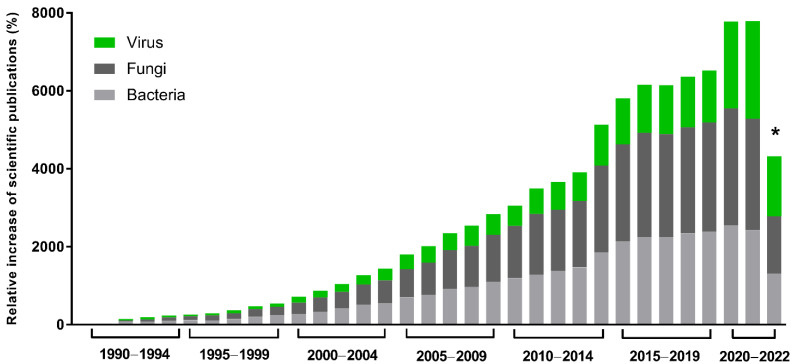
Scientific interest in microbial enzymes in biotechnology. Relative increase in scientific publications since 1990. The numbers were obtained from the Pubmed database using the names of the microbial groups (viruses, fungi, or bacteria), plus enzymes plus biotechnology. A total of 112,240 results were obtained. Relative increase was calculated by comparing the number of publications in a given year with the publication number in 1990, the first year of the historical record. * Data as of 12 September 2022.

**Figure 2 pathogens-11-01453-f002:**
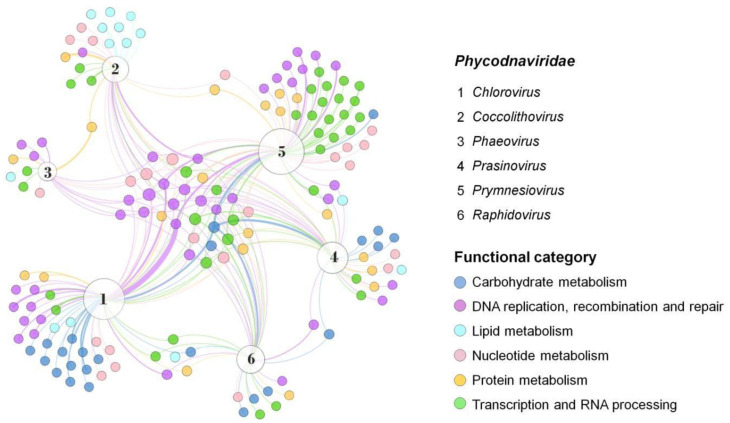
Presence and distribution of enzymes in the *Phycodnaviridae* family. Representatives of each genus were included and data on the diversity and abundance of enzymes grouped into different functional categories were obtained from genomic annotations publicly available on GenBank. A network graph was constructed using Gephi 0.9.7 using a force-based algorithm (ForceAtlas2), followed by a manual arrangement of nodes for better visualization. Node sizes are proportional to the degree of connection. The thickness of the edges is proportional to the number of genes of the same function in the genome of a virus. Virus representatives: (1) Paramecium bursaria chlorella virus 1; (2) Emiliania huxleyi virus 86; (3) Feldmannia species virus; (4) Ostreococcus tauri virus 5; (5) Phaeocystis globosa virus; (6) Heterosigma akashiwo virus 01. Full data are available in Appendix A.

**Figure 3 pathogens-11-01453-f003:**
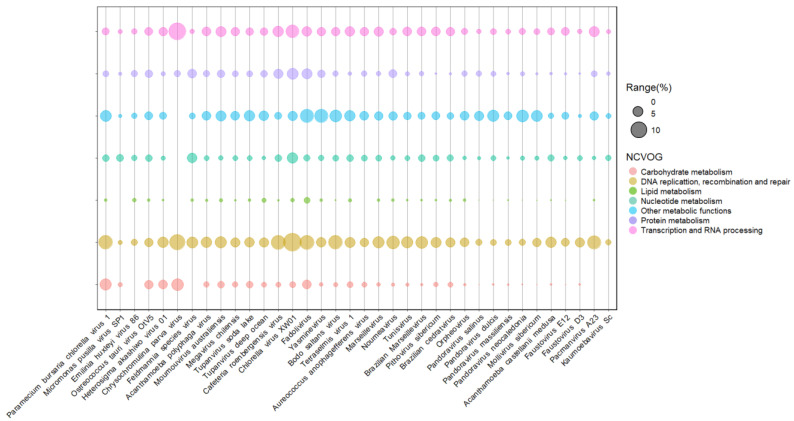
Presence and abundance of enzymes in giant viruses of protists. Bubble chart containing representatives of different viral families of the phylum nucleocytoviricota associated with algal and amoeba hosts. Bubble sizes are proportional to the number of enzymes represented as a gene percentage of a functional category in the virus genome. Data on enzyme diversity and abundance of each virus grouped into different functional categories were obtained from genomic annotations publicly available on GenBank and classified according to NCVOG categories.

**Figure 4 pathogens-11-01453-f004:**
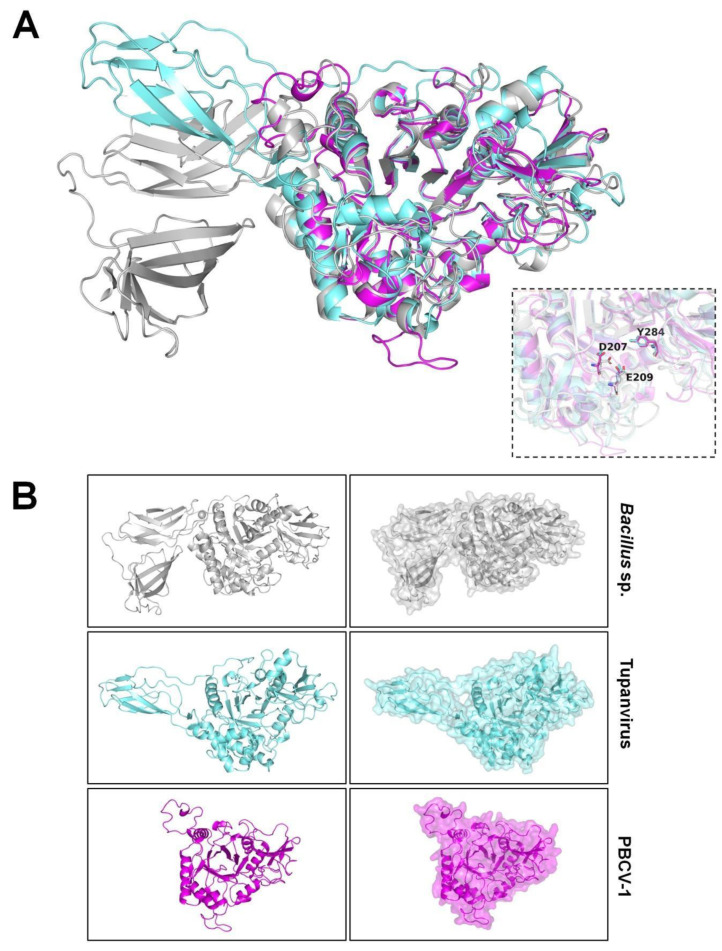
Models of chitinases from microbial sources. Modeled structures were obtained by SWISS-MODEL (https://swissmodel.expasy.org/ (accessed on 1 October 2022)), represented by drawing. (**A**) Three-dimensional alignment of chitinases from Bacillus cereus/Bacteria (gray), Tupanvirus/*Mimiviridae* (cyan), and Paramecium bursaria chlorella virus 1/*Phycodnaviridae* (pink). The alignment shows an equivalent core for all structures, with an evident difference in size, especially from B. cereus, with two-sided protein fragments, one being correspondent to Tupanvirus and absent in PBCV-1. The box in the right part of the image evidences key common residues at the active site of the enzymes (D207, E209, Y284); (**B**) Models from each microorganism represented as drawing and surface structures. The modeling of each target sequence was assessed with the best hit available (scored) after employing the search template and build model tools. Model alignment and analysis were assessed with the PyMOL software (v0.99c).

**Table 1 pathogens-11-01453-t001:** General genomic data of representatives of different groups of giant viruses of protists.

Family *	Genus *	Virus	Genome Length (bp)	GC%	CDS	Unclassified CDS #	NCBI Accession
*Phycodnaviridae*	*Chlorovirus*	Paramecium bursaria Chlorella virus 1	330,611	40.0	416	368	NC_000852.5
		Only Syngen Nebraska virus 5	327,147	42.4	357	204	NC_032001.1
		PBCV CVB-1	319,457	44.3	339	214	JX997160
		Acanthocystis turfacea chlorella virus 1	288,047	49.4	329	218	NC_008724.1
	*Coccolithovirus*	Emiliania huxleyi virus 86	407,339	40.2	472	409	AJ890364.1
	*Prasinovirus*	Micromonas pusilla virus SP1	173,451	40.6	242	220	NC_043129.1
		Ostreococcus tauri virus OtV5	186,713	44.8	268	208	NC_010191.2
	*Raphidovirus*	Heterosigma akashiwo virus 01	274,793	30.4	246	190	NC_038553.1
	*Prymnesiovirus*	Chrysochromulina parva virus	437,255	25.1	45	28	MH918795.1
	*Phaeovirus*	Feldmannia species virus	154,641	51.8	150	110	NC_011183.1
*Mimiviridae*	*Mimivirus*	Acanthamoeba polyphaga mimivirus	1,181,549	27.9	985	537	HQ336222.2
		Moumouvirus australiensis	1,098,000	25.1	899	438	MG807320.1
		Megavirus chilensis	1,246,130	25.3	1126	610	NC_016072.1
	*Tupanvirus*	Tupanvirus soda lake	1,516,267	29.0	1359	914	KY523104.2
		Tupanvirus deep ocean	1,439,510	29.4	1276	845	MF405918.2
	*Cafeteriavirus*	Cafeteria roenbergensis virus	617,453	23.3	544	376	NC_014637.1
		Chlorella virus XW01	407,612	21.9	200	90	OL828820.1
	*Klosneuvirus*	Klosneuvirus	1,573,080	28.6	1545	1017	KY684108.1
		Fadolivirus	1,573,504	27.1	1428	674	MT418680.1
		Yasminevirus	1,991,922	40.4	1434	926	UPSH01000001.1
	*Mesomimivirus*	Bodo saltans virus	1,385,870	25.3	1207	683	MF782455.1
		Tetraselmis virus 1	668,031	41.2	653	461	KY322437.1
		Aureococcus anophagefferens virus	370,920	28.9	384	309	OM876856.1
*Marseilleviridae*	*Marseillevirus*	Marseillevirus T19	368,454	44.7	428	273	NC_013756.1
		Noumeavirus	376,207	42.9	452	300	NC_033775.1
		Tunisvirus	380,011	43.0	484	355	NC_038511.1
		Brazilian Marseillevirus	362,276	43.3	491	347	NC_029692.1
*Pithoviridae*	*Pithovirus*	Pithovirus sibericum	610,033	35.8	467	339	NC_023423.1
	*Cedratvirus*	Cedratvirus A11	589,068	42.7	574	330	NC_032108.1
		Brazilian cedratvirus	460,038	42.9	533	325	LT994651.1
*Orpheoviridae*	*Orpheovirus*	Orpheovirus IHUMI	1,473,573	25.0	1199	753	NC_036594.1
*Pandoraviridae*	*Pandoravirus*	Pandoravirus salinus	2,476,870	61.7	1430	853	NC_022098.1
		Pandoravirus dulcis	1,908,520	63.7	1070	748	NC_021858.1
		Pandoravirus massiliensis	1,593,060	60.1	1269	1003	MZ384240.1
		Pandoravirus neocaledonia	2,003,190	60.6	1081	709	NC_037666.1
*Molliviridae*	*Mollivirus*	Mollivirus sibericum	651,523	60.1	523	424	NC_027867.1
		Mollivirus kamchatka	648,864	60.1	504	428	MN812837.1
*Medusaviridae*		Acanthamoeba castellanii medusavirus	381,277	61.7	470	358	AP018495.1
*Faustoviridae*	*Faustovirus*	Faustovirus E12	466,265	36.2	492	404	KJ614390.1
		Faustovirus D3	465,956	37.7	485	423	KU556803.1
	*Pacmanvirus*	Pacman A23	395,405	33.6	465	362	LT706986.1
	*Kaumoebavirus*	Kaumoebavirus Sc	350,731	43.7	429	391	NC_034249.1

* Only italicized taxa are currently defined by ICTV. # Genes with unknown functions based on NCVOG functional categories.

## Data Availability

All genomic data used in this work are publicly available at GenBank (https://www.ncbi.nlm.nih.gov/genbank/).

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
