# Peer review of "Giant Viruses as a Source of Novel Enzymes for Biotechnological Application"

_pathogens, 2022, doi:10.3390/pathogens11121453_

Round 1

Reviewer 1 Report

A comprehensive review about enzymes encoded by Phycodnaviruses and Amoeba giant viruses. Here are some questions.

First, Are there any studies about that viral metabolic enzymes were better or more efficient than bacterial or eukaryotic ones ?Because there were only several examples of commercialized viral enzyme involving DNA/RNA processing in this review. 

Secondly, is it possibly to provide the TM-score, which is used to evaluate the alignment between a query structure and known ones in the PDB library, between different chitinases ? 

Besides, were Alphafold or Robetta used for structure prediction of viral proteins except SWISS-model ? 

Thirdly, line 448-460: Why mention chlorovirus, which was algal virus and not grouped into extended mimiviridae, in the section of amoebal virus suddenly ? 

Reviewer 2 Report

This review presents the promise of giant viruses as a new source of enzymes for biotechnological applications. It is an interesting study and a clear review of the potential of giant viruses. Nevertheless, some point should be more developed.

1- Globally for some viruses, the authors only mention the interesting enzymes present in their genomes without discussing their potential use in biotechnology. It would be relevant to try to develop more about it for Phaeoviruses, Prymnesioviruses.

2- Chloroviruses:

- Line 262-270: The paragraph referring to hyaluronan synthase and xanthine synthase genes.

It would be interesting to discuss their possible use in biotechnology rather than their role in the virus life cycle.

3- Giant viruses of amoebae

- Line 358: “The mimiviruses translation-related genes set includes many aminoacyl-tRNA synthetases, transfer RNAs (tRNAs), and translation factors [101–103].”

Line 369: “These viruses have the most complete protein translation apparatus of the virosphere to date, which includes up to 70 types of tRNAs, and factors related to tRNA maturation and stabilization besides all the 20 aminoacyl-tRNA synthetases [105].”

Authors describe for mimiviruses and tupanviruses, the presence of enzymes involved in the translation. It would be interesting to develop if these enzymes could be employed in biotechnology?

- Line 372: Authors mention that tupanvirus possess a citrate synthase. Could they discuss how this enzyme can be worthwhile in biotechnology field?

- Lines 419-428: In this paragraph, there a description of marseilleviruses.

Authors indicate that those viruses had histone-like proteins. They are not the only ones, medusaviruses and clandestinovirus also possess histone-like proteins. It would be helpful if the authors could discuss how these proteins can be relevant.

As phycodnaviruses, marseilleviruses and other giant viruses also had methyltransferases (Jeudy et al, 2021). Why the authors do not develop more about them?

Don't marseilleviruses have other proteins of potential interest besides histones and methyltransferases?

- Lines 461-471: For the faustoviruses, kaumoebavirus and pacmanvirus, the authors only mention the mRNA capping enzyme.

Are there any other enzymes that could be interesting to application in biotechnology?

- Lines 472-482: the paragraph on Yaravirus, I would rather put it in the conclusion to discuss the huge potential of all the giant viruses as source for enzymes with the large proportion of genes with unknown functions in their genomes highlighted by Yaravirus with 90% of its genome unknown.

- Line 418: “Besides the Mimiviridae family, several other groups of large and giant viruses of amoeba are proposed to be part of the Nucleocytoviricota phylum, such as Marseilleviruses, pandoraviruses, molliviruses, pithoviruses, cedratviruses, Faustoviruses, kaumoebaviruses, and orpheoviruses, among others [15].”

Authors mention molliviruses, pithoviruses, cedratviruses and orpheoviruses. Is there any reason not to have included a description of these viruses?

4- Conclusion

- Line 493: Authors mention metagenomics. I would suggest to develop more about it like it is a sequencing more and more used in the giant viruses’ field. Moreover, in the literature, some studies described unusual proteins found in giant viruses metagenomic experiments.

- Line 495: “Furthermore, many studies argue that aquatic viruses represent the greatest unexplored genetic diversity on the planet [17,139–142].”

In addition to aquatic environments, giant viruses have been discovered in the permafrost. This environment with climate change will probably allow new discoveries. This would be interesting to add to the conclusion.

5- Figures

- Figure 1 and 3: the writing looks pixelated. It should require a little improvement of the picture quality

- Figure 4: “cartoon” might be replaced by “drawing”

6– Minor revisions.

Line 91: “Chlorovirus” should to be in italic.

Line 98: “Curiously, in a genomic characterization of the EhV-86, of the 472 coding sequencing (CDS) regions, only 63 genes had a known function so far (Table 1)”.

Need to be rephrase, it probably misses words.

Line 531-533: Reference 2 and 3 are identical.
